# Electronic and Optical Properties of Two-Dimensional Tellurene: From First-Principles Calculations

**DOI:** 10.3390/nano9081075

**Published:** 2019-07-26

**Authors:** David K. Sang, Bo Wen, Shan Gao, Yonghong Zeng, Fanxu Meng, Zhinan Guo, Han Zhang

**Affiliations:** Shenzhen Engineering Laboratory of Phosphorene and Optoelectronics, Collaborative Laboratory of 2D Materials for Optoelectronics Science and Technology, Engineering Technology Research Center for 2D Material Information Function Devices and Systems of Guangdong Province, Shenzhen University, Shenzhen 518060, China

**Keywords:** few-layer Tellurene, density functional theory, spin orbital coupling, quantum confinement effect

## Abstract

Tellurene is a new-emerging two-dimensional anisotropic semiconductor, with fascinating electric and optical properties that differ dramatically from the bulk counterpart. In this work, the layer dependent electronic and optical properties of few-layer Tellurene has been calculated with the density functional theory (DFT). It shows that the band gap of the Tellurene changes from direct to indirect when layer number changes from monolayer (1 L) to few-layers (2 L–6 L) due to structural reconstruction. Tellurene also has an energy gap that can be tuned from 1.0 eV (1 L) to 0.3 eV (6 L). Furthermore, due to the interplay of spin–orbit coupling (SOC) and disappearance of inversion symmetry in odd-numbered layer structures resulting in the anisotropic SOC splitting, the decrease of the band gap with an increasing layer number is not monotonic but rather shows an odd-even quantum confinement effect. The optical results in Tellurene are layer dependent and different in *E* ⊥ C and *E* || C directions. The correlations between the structure, the electronic and optical properties of the Tellurene have been identified. Despite the weak nature of interlayer forces in their structure, their electronic and optical properties are highly dependent on the number of layers and highly anisotropic. These results are essential in the realization of its full potential and recommended for experimental exploration.

## 1. Introduction

Since the advent of graphene, much effort has been devoted to the search of two-dimensional (2D) layered materials beyond graphene which can be obtained from the layered van der Waals (vdW) solids [1]. Due to the naturally terminated surface with vdW interactions rather than dangling bonds [2,3], and the out-of-plane bonds, 2D materials are stable in ambient conditions, and thus offer them advantages with extraordinary mechanical, electrical, and optical properties, which are different from their counterparts in bulk materials. Atomic layers of transitional metal dichalcogenides [4,5,6,7,8] and the metal chalcogenides such as MoS_2_, GaTe and InSe [9,10,11,12,13,14,15] have gained a great deal of attention due to their unique properties which are tunable via layer control. InSe have shown strong layer dependent band gap value changing from 2.8 eV for monolayer to 1.3 eV for thick film, via luminescence [16], and magnetoluminescence [17] studies. Moreover, their optical response differs greatly from their bulk counterparts, and has shown excellent electroluminescence [18], high photoresponsivity [19] with a broad spectral response [15,20], and band gap tunability [13].

Recently, the efforts on exploiting the application of 2D anisotropic vdW solid semiconductor materials in electronic and optoelectronic field have been on the rise [21,22,23]. One of the most studied 2D anisotropic materials in recent years is the black phosphorus (BP) and has gained a lot of interest, which shows strong angle-dependent Raman responses and anisotropic thermal conductivity [24,25,26]. In addition to BP, another anisotropic 2D material, Tellurium, has come to us owing to its unique chained structures, fascinating physical properties and potential applications [27]. Bulk Tellurium is a *p*-type elemental helical semiconductor with a narrow band gap (0.35 eV) [28] at room temperature. It exhibits excellent physical properties such as photoconductivity [29], nonlinear optical response [30], and high thermoelectric [31] and piezoelectric [32,33] properties. These properties make bulk Tellurium a potential candidate for applications in photoconductors [34,35], solar cells [36], infrared acousto-optic deflectors [37], field effect transistors [38], self-developing holographic recording devices [39], radiative cooling devices [40], topological insulators [41] and gas sensing [42]. However, when the size of Tellurium is in nanoscale, it has been confirmed to exhibit some new physical properties [43,44,45,46,47]. In the last few years, a successful fabrication of Tellurium nanostructures has been witnessed such as ultrathin Te-epilayers [43], nanowires [44], nanoplates [45], and nanorods [46]. Recently, a breakthrough in obtaining a solution–grown 2D Tellurene (Te, Tellurium in few-layer structure) with a large area, and high-quality flake with high mobility of ~700 cm^2^ has been reported in 2018 [47]. This confirmed the existence of Tellurene via experiments and indicated a new member of the 2D materials’ family from the main group—VIA. With this new development, not only has a new twist in chiral-based semiconductor materials been opened but also renewed interest towards the newfound 2D Tellurene. However, studies on monolayer and few-layer 2D-Te from the main group-VIA are conducted seldomly and, therefore, novel electronic and optoelectronic properties of 2D chiral vdW solid mono-elemental material Tellurene are still in their infancy with many remarkable phenomena to be discovered.

Our interest is to focus on new electronic structure and optical properties of 2D Te, and the potential use in optoelectronic and nanoelectronic fields. In this work, spin–orbit coupling (SOC) effects were considered in the calculation of the electronic structures of bulk Tellurium, few-layer and monolayer Tellurene. Due to underestimation of band gap energy by Generalized Gradient Approximation (GGA-PBE) functionals, only a monolayer was computed under the HSE06 functional for the purpose of comparison, because of the high computational demand from a Heyd–Scuserian–Ernzerhorf (HSE06) hybrid functional. Moreover, the optical properties of Tellurene can be determined qualitatively and quantitatively via dielectric function, which shows a strong dependence on layer thickness and polarization directions. In this work, electronic structures, optical and work function are the fundamental properties that are excited-state quantities influenced by many-electron effects, and thus prompted the selection of density functional theory (DFT) calculations that incorporates self-energy corrections and excitonic effects. 

## 2. Methods

The geometry structure of 2D Te was optimized by using density functional theory (DFT) implemented in the Vienna Ab-initio simulation package (VASP) [48,49]. The exchange correlation functional under the generalized of the Perdew–Burke–Ernzerhof (PBE) scheme [50] and the projector augmented wave (PAW) [51] type Pseudopotentials was employed. The Heyd–Scuserian–Ernzerhorf (HSE06) hybrid functional [52] was used in order to obtain an accurate band gap in monolayer (1 L) and bulk Te because this functional is computational and machine demanding. The cut off energy plane-wave basis was set to be 500 eV. The vacuum space vertically along the *z*-direction was set to over 16 Å in order to mimic the 2D system and to avoid the interaction between layers and its periodic images. During optimization and single point calculations, vdW interaction is considered in PBE exchange functional proposed by Grimme (DFT-D2) [50] to account for the inter-layers’ interaction in few-layer Te (layer number (L) > 1). The atomic positions were fully relaxed until the forces on all atoms were less than 10^−2^ eV/Å. The energy convergence criterion was set to be 10^−6^ eV/cell using the Fast (Davison and RMM-DIIS) algorithm. A dense k-mesh of 15 × 15 × 1 sampling with the Monkhorse–Pack scheme [53] was used for both optimization and single point calculation in the Brillouin zone integration.

The Phonon property of monolayer Te was calculated using the Full Brillouin zone method implemented in Phonopy [54]. A 4 × 4 × 1 supercell was constructed to calculate the atomic forces employing the VASP 5.4, using the Fast (Davison and RMM-DIIS) algorithm with electronic convergence set to 10^−7^ eV. To confirm the thermal stability of monolayer Tellurene, an Ab-initio molecular dynamics (AIMD) simulations within the framework of canonical ensemble or NVT ensemble (constant number of particles, volume and temperature) was performed [55]. To observe changes in monolayer Te at the atomic level in the present equilibrium state and reduce the periodic boundary constraints environments, a cell with same length, a = b in the *x* and *y* direction with 4 × 4 × 1 (48 atoms) was considered. Sampling configuration space was carried out at room temperature. The valence electrons from 4d^10^5s^2^5p^4^ orbitals of Te are considered during calculations.

## 3. Results and Discussion

### 3.1. Crystal Structure

The trigonal structure of the *β*-Tellurene space group-P3_1_21-$D_{3}^{4}$ consisting of a helical-chains arranged in a hexagonal array which spirals around axes parallel to the *x*-axis and exhibits a highly anisotropic growth tendency which makes it possible to synthesize by controlling the experimental conditions. This singularity is qualitatively attributed to the structure reconstruction of Te as shown in Appendix A. As shown in Figure 1a, the proposed model structure of Tellurene is composed of planar four-membered and chair-like six-membered rings alternately arranged in a 2D lattice. The geometry lattices of the optimized unit cell are symmetrized as a = 5.90 Å and b = 4.46 Å, respectively, which are in good agreement with the lattices from the experimental results [55]. The Tellurene structure exhibits two different orientations within its basal plane i.e., the *x*-direction (or armchair edge) parallel to the chain and the *y*-direction (zigzag edge) normal to the chain. The unit building block of Te consists of -Te-Te-Te- helical chains arranged in a parallel trigonal lattice (Figure 1b). It is terminated by Te-lone pair of orbitals, which constitute the electronic states at the valence-band maximum (VBM). In the 2D Tellurene lattice, the atoms are arranged in helical chains at the center and corners of regular hexagonal and the atoms interaction in 2D-Te is weaker than in β-bulk Tellurium and this is attributed to the structural reconstruction from bulk to few-layer.

As shown in Figure 1c, the Tellurene layer has a chiral chain along the [100] direction pointing the *x*-axis, which demonstrates that the Te has a strong tendency to grow along the *x*-axis totally opposite to bulk trigonal *β*-Tellerium, which exhibits a tendency of growth in the direction [001] [55,56]. The highly anisotropic crystal structure of Te displays covalent bonded helical chains which are governed by vdW interaction, forming a trigonal lattice. This intrinsic anisotropy of few-layer Te shows unique exceptional properties such as high mobilities at room temperature, direct semiconductors at a limit of the monolayer layer, and a layer dependent band gap. The intra- and interlayer coupling within layers are covalently bonded and strongly held by vdW forces respectively, while the interactions of layer-to-layer are predominantly governed by vdW forces of attraction and are much weaker, which demonstrates that Te possesses a unique layered structure different from the other known 2D layered materials.

### 3.2. Electronic Band Structures of Few-Layer Tellurene

Band gap is a very important characteristic because it influences the electrical and optical properties of the 2D material due to the manifestation of quantum size effects as a result of a reduced dimension. The electronic band structure of layered crystals materials depends on the composition, thickness and stacking order of the component’s layers. Monolayer Te shows a direct band gap at Γof the Brillouin Zone (BZ), where the optical transition is determined accurately and whose joint density of states (JDOS) have a van Hove singularity (Figure 2). Monolayer Te is a direct band semiconductor with a band gap value of 1.0 eV. The gap value of bilayer Te reduced to 0.9 eV with an indirect band gap where the VBM is located between A-B and conduction band minimum (CBM) is between Γ-B k-points of the BZ (Figure 2). Tellurene has shown to have ABAB stacking arrangement and the transformation from slightly indirect to direct semiconductor is likely to be linked to arrangement of layer component and the thickness control. Due to quantum confinement along the normal-plane direction of Te crystal, the band gap energies show odd-even-layer-dependent band gap characteristics as shown in Appendix A. Moreover, the band structure exhibits strong anisotropic energy wavevector momentum dispersion. The disappearance of interlayer vdW interactions in monolayer Te may play an important role towards the electronic band structure from few-layer to monolayer Te. This great transition is demonstrated in the electronic structure of bilayer Tellurene.

Band gap values reduced continuously upon addition of consecutive layers onto monolayer Te and reach 0.30 eV for six-layered Tellurene. The band gap value-thickness dependence obeys the exponential decay law, which is 1.0 eV (1 L), 0.9 eV (2 L), 0.6 eV (3 L), 0.5 eV (4 L), 0.4 eV (5 L) and 0.3 eV (6 L), respectively. Since the PBE functional is known to underestimate the band gap, hence, the HSE06 calculations were performed to extract a more accurate band gap in monolayer and bulk Te only because of the computational demand from HSE06 functional. The electronic band structure obtained from HSE06 is shown in Appendix A with gap energy of 1.5 eV. It is noted that, from 3 L to 6 L, the VBM and CBM are located between A and B points of the BZ. However, their exact points are layer dependent, as shown in the Appendix A. In few-layer Te system, the dispersion of VB and CB is large and hence small band gaps may be attributed to the strong interlayer interaction and strong coupling interactions.

The bulk Tellurium exhibits a direct band gap of 0.32 eV and 0.052 eV from HSE06 and GGA-PBE functional, respectively, as shown in Appendix A. In this communication, the Te-material has been confirmed to exhibit adjustable band gap values from monolayer, few-layer to bulk. From our results, the monolayer and few-layer Tellurene depict small band gaps which allow for photo-stimulated alteration of electron transfer using visible and infrared light source to excite the band-edge electron transition with a long wavelength light source as compared to other materials with wide band gaps. The calculated band structures are reasonable with the infrared absorption data in view of high optimization of the crystal potential, which gives a reasonably accurate prediction of the band gap energy [22,43,44,56,57,58].

To understand the contribution of different orbitals to the electronic states in Te, total density of states (TDOS) and projected density of states (PDOS) were calculated based on PBE functional and presented in Figure 3. For the partial density of state (PDOS), both the electronic locations (VBM and CBM) are dominated by the *p* orbitals and this is an iconic qualitative feature of electronic states of Te. The calculated PDOS shows a considerable *s*-*d* mixing in the valence band and conduction band states and the mixing level is strongly dependent on the state location. The presences of mixed states do not alter the qualitative features of the band gap structure in the absence of spin–orbital coupling (SOC); thus, the influence of mixing effects is much smaller on the band structures than the SOC splitting. The DOS spectrum exhibits odd-even quantum confinement. The shift in valence band maximum towards the Fermi level is not gradual but shows odd-even quantum confinement character. In odd numbered layered structures, the spin–orbit coupling and lack of inversion symmetry are attributed to this phenomenon.

Furthermore, the phonon dispersions exhibit no imaginary frequencies, indicating that the monolayer Te is dynamically stable as shown in Appendix A, while the AIMD as shown in Appendix Ab showed that monolayer Te is thermally stable at 300 K, thus suitable for optoelectronic devices operating optimally at room temperature.

### 3.3. Optical Properties

The electronic structure of few-layer Te and the resulting optical characteristics emanate from the *p*-orbital, which is highly pronounced in the conduction and valence bands of the Te. It leads to engineering novel optical behaviors of Te, which are not exhibited in *sp*-hybridized materials, and makes Te a new candidate for optoelectronic and electronic applications. Investigating the optical properties of 2D-Tellurene is important for the optoelectronic applications. A complex dielectric function (Equation (1)), which is the combination of the imaginary and real parts of the permittivity, can be used to investigate the dielectric properties of the few-layer Te. This complex function characterizes the responses from the material upon interacting with the incident polarized light:
*ε* (*ω*) = *ε*_1_(*ω*) + *ίε*_2_(*ω*),(1)
where *ίε*_2_(*ω*) is the imaginary part and *ε*_1_(*ω*) is the real part of the dielectric function. The imaginary part and real part were derived from the Kramers–Krӧning relationship within the relationship framework of random phase approximation (RPA) method considering the Fermi–Dirac distribution function. The imaginary part of the dielectric function is given by the equation:(2)$$ί\varepsilon_{2\;}\left(\omega \right)\;=\;\frac{4\pi e^{2}}{m^{2}\omega^{2}}\sum_{c,v}\int_{B}^{Z}d^{3}k|\langle v_{k\;}|p_{2}|c_{k}\rangle {|}^{2}\times \delta \;(E_{ck}\;\;-\;E_{vk}\;-\;\hbar \omega ),$$where the term $\langle v_{k\;}|p_{2}|c_{k}\rangle$ consists of the unoccupied and occupied states of electrons in the valence and conduction bands. The *e, m,* and ℏω are the electron charge, mass and photon energy, respectively. The ($ί\varepsilon_{2\;}\left(\omega \right)$) is the imaginary function where $\varepsilon_{xx}\neq \varepsilon_{yy}$ and fundamental value $\;p_{2}$, can be substituted with $\varepsilon_{xx}$ and $\varepsilon_{yy}$, which can take the form $\;p_{xx}$ and $\;p_{yy}$. The intraband transitions of the electrons are the main part of the optical spectra identifier in semiconductors, which can be gained by the imaginary part of the dielectric function. Moreover, the polarization of the electric field (*E*) of the incident photon and the crystal symmetry (*C*) are significant factors in evaluating the optical behavior of the material, and all the optical functions and spectra are determined along the *E* ⊥ *C*(x) and *E* || *C*(y) directions of polarization. The imaginary spectra of monolayer, few-layer Te and bulk Tellerium are presented in Figure 4. The imaginary part of the dielectric function relates to photon absorption. The optical energy gaps along *E* ⊥ C and *E* || C in polarization directions exhibit no differences but only shift to higher energy values as the layer thickness is reduced to a limit of monolayer. In addition, the spectra dispersion in both directions is not regular, and this is attributed to the anisotropic nature of Te crystal.

The real part of the dielectric function is given by Equation (3), which is derived from the framework of the RPA method:(3)$$\varepsilon_{1}(\omega )\;=\;1\;+\;\frac{2}{\pi }p\int_{0}^{\infty }\frac{ώ\varepsilon_{2}\left(ώ\right)}{{ώ}^{2}-\omega^{2}}\partial ώ,$$where *p* is the fundamental value of this function, and *ίε*_2_(*ω*) and *ε*_1_(*ω*) are the imaginary part real part of the dielectric function, respectively. The real part of the dielectric function is presented in Figure 5. The static dielectric constants provided by the real part of the dielectric function at 0 eV. In both directions, the static dielectric constants exhibit layer dependence. The static dielectric constants are 5.9 (1 L), 10.3 (2 L), 14.3 (3 L), 20.0 (4 L), 21.4 (5 L), 27.3 (6 L) and 38.3 (bulk) in *E* ⊥ C, and 6.9 (1 L), 9.3 (2 L), 11.2 (3 L), 15.9 (4 L), 16.8 (5 L), 20.2 (6 L) and 53.5 (bulk) in *E* || C directions, respectively. The static dielectric constants increase with the increasing number of layer and this is attributed to the interplay of vdW force. From the low energy region, the static dielectric constant increases with increasing layer number and reaches maximum values of 11.4 (4.5 eV), 16.4 (3.3 eV), 21.6 (3.0 eV), 30.6 (2.7 eV), 31.3 (2.6 eV) 40.1 (2.5 eV) and 43.8 (2.3 eV) in *E* ⊥ C, and 15.9 (4.5 eV), 15.0 (3.3 eV), 15.3 (2.4 eV), 20.5 (2.4 eV), 20.5 (2.4 eV), 25.1 (2.4 eV) and 77.2 (2.6 eV) in *E* || C, respectively. The curves in both orientations are not regular and this is due to the anisotropic structure of the Tellerium crystal. 

The reflectivity spectra of monolayer, few-layer and bulk in both orientations are plotted in Figure 6. The features in these spectra correspond to the real part of the complex dielectric function. In both orientations, Te exhibits small reflectivity, thus demonstrating that the monolayer, few-layer Te and bulk Tellerium have nearly transparent features. Moreover, monolayer and few-layer exhibit almost the same behavior under the polarized light in both orientations. 

The optical absorption spectrum of monolayer, few-layer Te and bulk Tellerium obtained by means of DFT calculations is presented in Figure 7. The absorption spectrum of Te shows that absorption depends largely on the polarization of the incident radiation, and the absorption commences at the lower frequency for radiation polarized with the electronic vector perpendicular than the parallel to the trigonal axis. The absorption energy increases as the number of layers decreases in the range of 0–3 eV. As shown in Figure 7 of the optical absorption spectrum, two peaks can be seen in the monolayer. The one at 7.3 eV is attributed to the direct allowed transition from the valence band (*p*-bonding triplet VB1) to conduction band (*p*-anti-bonding triplet CB1), and another at 5.5 eV is attributed to the forbidden direct transition from the valence band (*p*-lone pair VB2) to the conduction band (*p*-anti-bonding triplet CB1). The bulk Tellurium shows a broad absorption peak attributed to the direct transition located at approximately 5.5 eV in *E* ⊥ C and 4.5 eV in *E* || C direction while the peaks of few-layer and monolayer shift to higher energies. In the *E* ⊥ C polarization direction, the enhanced optical interband relaxation rate in the monolayer is attributed to the increase of electronic relaxation time, and this strongly suggests a substantial change of the electronic structure from few-layer to monolayer.

Generally, the absorption in both orientations is large and indicates strong photon–matter interaction in the entire wavelength range. The positions of the lower-energy absorption peaks of the two directions (normal and parallel) are not the same and exhibit the strong anisotropic characteristic of the monolayer, few-layer Te and bulk Tellurium. There is a sharp increase in the extinction coefficient in both directions and this is a signature of the anisotropic nature of the band structure. Light interacts more strongly with the Te along the *y*-direction (normal to the chain) as seen in the peak intensities, and this shows strong dependence on the crystalline orientation and thickness.

The calculated index of refraction from monolayer layer (1 L) to few-layer (2 L–6 L) Te is presented in Figure 8. The refractive index is a dimensionless number which describes the behavior of the light propagating through a medium. This phenomenon is gaining attention in 2D materials due to its application in optoelectronic devices. The refractive indices in *E* ⊥ C(n_o_) and *E* || C(n_e_) polarization directions are 4.11 (1 L), 3.99 (2 L), 4.05 (3 L), 4.68 (4 L), 4.73 (5 L), 5.24 (6 L) and 3.53 (1 L), 4.13 (2 L), 4.76 (3 L), 5.60 (4 L), 5.71 (5 L) and 6.48 (6 L), respectively. The birefringence (*Δn* = *n*_e_ − *n*_o_) is an optical property, quantified as the differences between the reactive indices exhibited by the material in both polarization directions. The calculated birefringence of monolayer Te was established to be −0.58, indicating that this crystal possesses negative optic axis, like bulk Tellurium, which exhibits a refractive index of ~ 9.06 and ~ 7.00 in *E* ⊥ *C*(n_o_) and *E* || *C*(n_e_) polarization directions, respectively, yielding a negative birefringence of −2.06 as shown in Appendix A, while the refractive index of few-layer Te with layer numbers of 2 L, 3 L, 4 L, and 6 L are 0.14, 0.71, 0.92, 0.98 and 1.24, respectively. Few-layer Te exhibits positive birefringence, showing that they are principally positive crystals. It is noted that the refractive index in the *E* || C(n_e_) polarization direction increases with an increase in the number of layer thickness from monolayer, few-layer to bulk. Tellurene clearly demonstrates to be optically anisotropic material suitable for designing the cross-polarizing filters in optical microscopes.

Research on work function, which is described as minimum energy to remove an electron from the surface of the system, should benefit from the advances of the 2D materials. The energies are connected by the equation *h v* = *φ* + *E*_k_, where *E*_k_ is the maximum kinetic energy of the emitted electrons in joules (J). The energy of the photon of light (*h v*), where *h* is the Plank constant and *v* is the frequency, and the work function (*φ*) are the minimum energies needed to eject an electron from the surface of the material. When *E*_k_ is 0 eV, the photon energy is equal to the work function, and the threshold frequency is attained as *v**_o_*, and the equation becomes *h v_o_* = *φ*. Therefore, the threshold frequency is the minimum frequency of the incident light to initiate the photoemission process. Work function can be influenced by the variation of layer thickness. As shown in Figure 9, tuning of layers has a strong effect on work functions of 2D-Te and hence can be a powerful tool in controlling the work functions. The calculated threshold frequencies (*v**_o_*,) are 1.03 × 10^15^ Hz (1 L) 1.03 × 10^15^ Hz (2 L), 1.04 × 10^15^ Hz (3 L), 1.05 × 10^15^ Hz (4 L),1.06 × 10^15^ Hz (5 L) and 1.07 × 10^15^ Hz (bulk), respectively. When the thickness reduced from bulk to monolayer, the work function of the 2D-Te decreases monotonically from 4.44 eV to 4.25 eV, and this is attributed to the quantum size effect existing in this atomically thin Te. Work functions show smooth changes, and this gives insight on the proper selection of appropriate layer thickness for the designing of photoemission and field emission nanodevices.

## 4. Conclusions

In summary, the study reveals a surprising emergence of electronic structure and optical properties of monolayer and few-layer Te. The density functional theory and many-body perturbation theory show that 2 L–6 L is an indirect band gap, due to structural reconstruction, which leads to crossover to a direct band structure in the limit of the monolayer. The 2D-Te has an energy gap that can be tuned from 1.0 eV for the monolayer to 0.3 eV for the six-layer structure. These predictions of indirect to direct band gap transitions from multilayer to monolayer Te are fascinating behaviors arising from *p*-orbital related interactions in the Te system. The synergic combination of a thickness dependent narrow band gap and optical properties fully demonstrates Te as a highly promising candidate for fast and broadband photo-detection and photocell solar applications. The optical spectra in *E* ⊥ C and *E* || C directions show differences, which change the threshold of photo-absorption energy, real and imaginary of the dielectric function as a result of layer control. As a narrow band gap semiconductor, *p*-type-2D-Te enjoys plenty of advantages in future applications as building blocks for functional optoelectronic devices. The dramatic changes in electronic band structure and excellent absorption are foreseen to pave the way not only for ultrathin-high speed transistors but also optoelectronic devices working optimally in the near-IR, UV and Visible regions and miniaturization of optoelectronics and electronic devices. This work gives insight on understanding the layer dependence band gap and anisotropic light–matter interaction in 2D-Te as well as providing a brilliant guideline for the exploration of applications in the electronic and optoelectronic devices based on 2D-Te.

## Figures and Tables

**Figure 1 nanomaterials-09-01075-f001:**
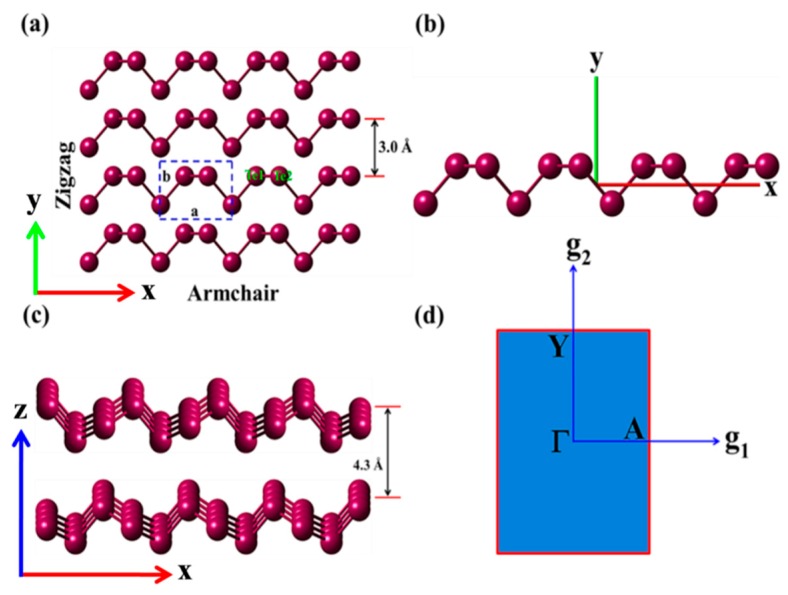
Structural model of trigonal crystal of 2D-Tellurene. (**a**) top view of Tellerene showing armchair and zigzag orientations and the enclosed section with dotted line shows the unit cell with four-sided dimensions; (**b**) single helical chain of 2D-Tellurene growing along the *x*-direction [100]; (**c**) two-layer chiral-helical-chains vdW structure; (**d**) the first Brillouin zone of the conventional unit cell of 2D-Te in all calculations of the monolayer and few-layer of Tellurene.

**Figure 2 nanomaterials-09-01075-f002:**
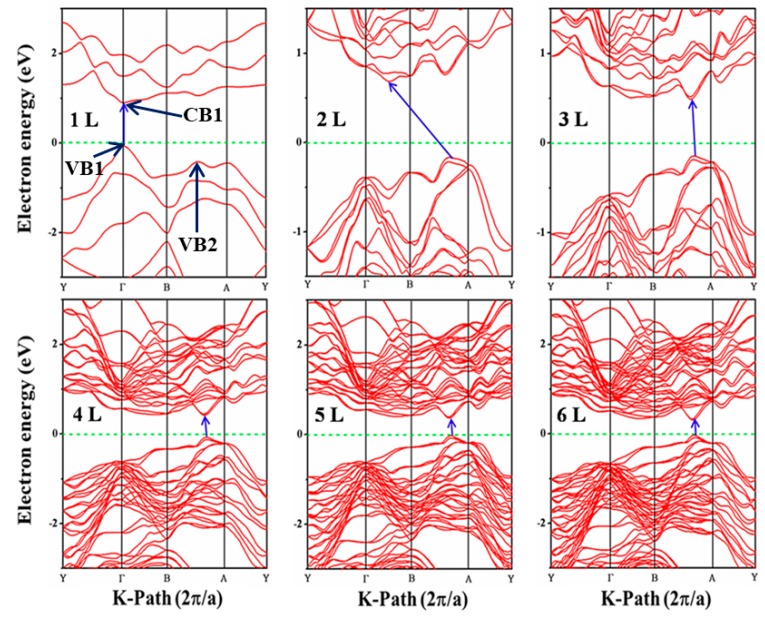
Electronic band structure extracted from PBE with SOC for monolayer (1 L) and few-layer (2 L–6 L) Tellurene.

**Figure 3 nanomaterials-09-01075-f003:**
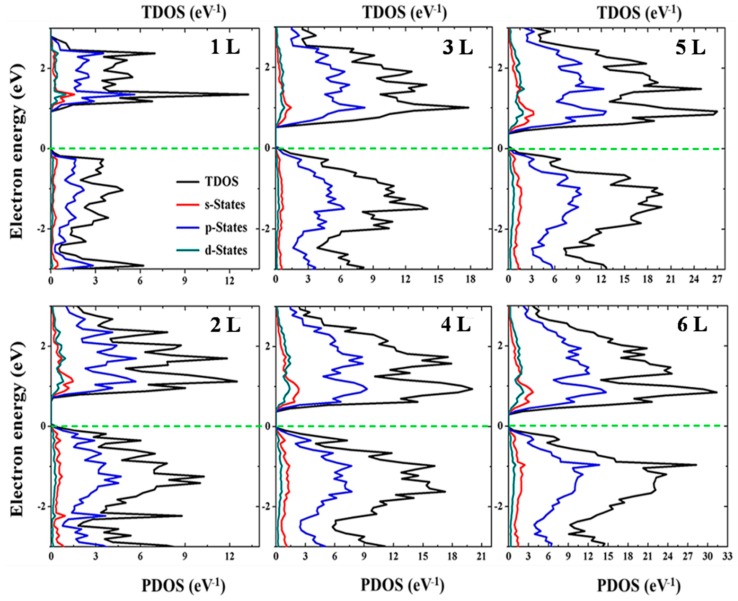
Total Density of States (TDOS) and Partial density of state (PDOS) of monolayer (1 L) and few-layer (2 L–6 L) Tellurene obtained from the PBE functional with spin orbital couple (SOC).

**Figure 4 nanomaterials-09-01075-f004:**
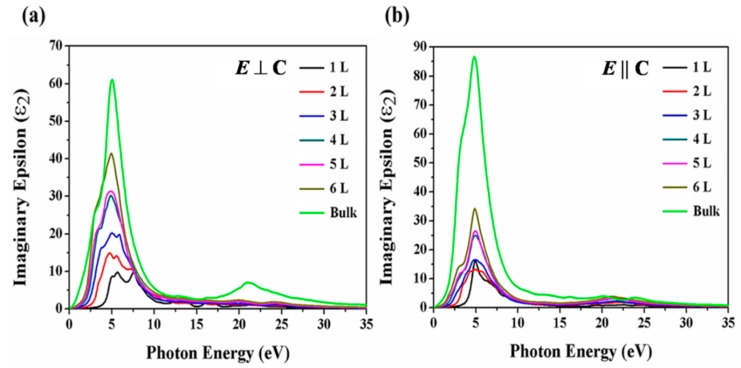
Calculated imaginary part of the dielectric function along (**a**) *E* ⊥ C and (**b**) *E* || C polarization directions of the few-layer Te and bulk Tellerium.

**Figure 5 nanomaterials-09-01075-f005:**
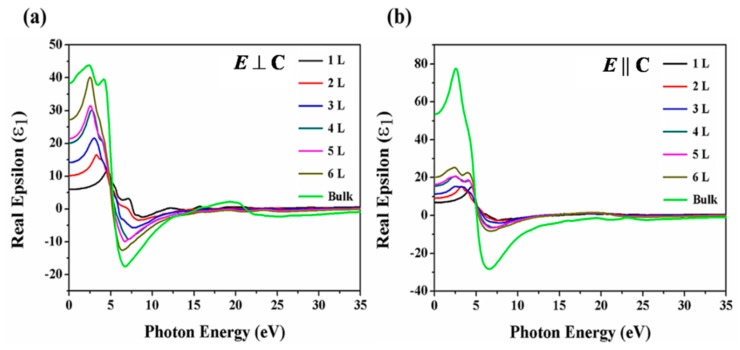
Calculated real part of the dielectric function along (**a**) *E* ⊥ C and (**b**) *E* || C polarization directions of the few-layer Te and bulk Tellerium.

**Figure 6 nanomaterials-09-01075-f006:**
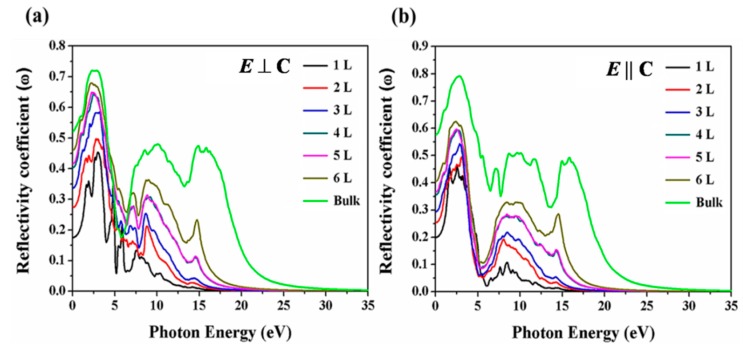
Calculated reflectivity coefficient along (**a**) *E* ⊥ C and (**b**) *E* || C polarization directions of the few-layer Te and bulk Tellerium.

**Figure 7 nanomaterials-09-01075-f007:**
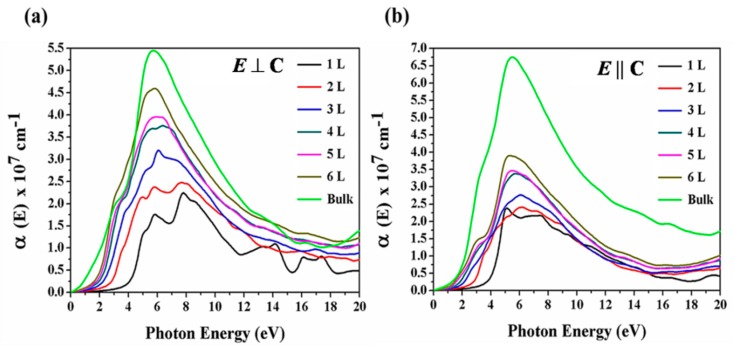
Calculated absorption spectra along (**a**) *E* ⊥ C and (**b**) *E* || C polarization directions of the few-layer Te and bulk Tellerium.

**Figure 8 nanomaterials-09-01075-f008:**
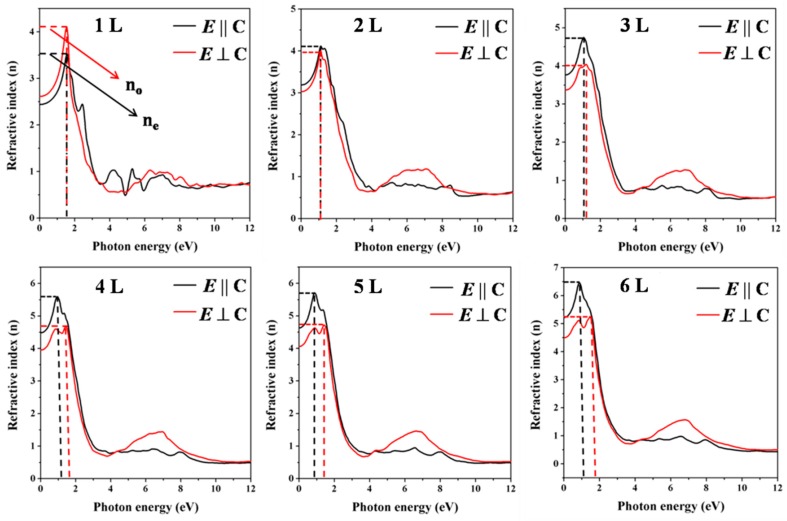
Calculated refractive index along *E* ⊥ C (n_o_) and *E* || C (n_e_) polarization directions of the few-layer Tellurenes.

**Figure 9 nanomaterials-09-01075-f009:**
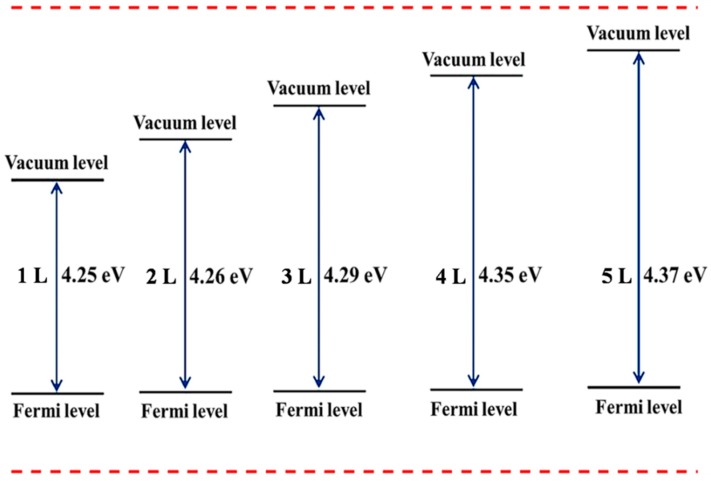
The work function schematic of monolayer and few-layer Te.

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
