# Peer review of "Electronic and Optical Properties of Two-Dimensional Tellurene: From First-Principles Calculations"

_nanomaterials, 2019, doi:10.3390/nano9081075_

Round 1
Reviewer 1 Report
The manuscript by Sang et al. reports the results on electronic structure, structural reconstruction and stability of several tellurenes – the slabs of tellur of different thickness. The subject of this manuscript belongs to the topics of Nanomaterials. The Authors have employed two calculational schemes of the density-functional theory (DFT) to reproduce correctly the band gap of single-layer tellurene and to discus the trends in evolution of the band gap for other tellurenes as a function of thickness. The study is performed on a high computational level, the results seem to be trustworthy and the Authors have clearly reported the results. I recommend this manuscript for publication in Nanomaterials after minor revision, according to following remarks.
1) Lines 162-163, the Authors wrote: “Due to quantum confinement along the normal-plane direction, Te shows odd-even-layer-dependent band gap characteristic in Figure S2b…” Honestly, I don’t see anything here, looking as an odd-even-layer-dependence. The plot has a shoulder between double- and triple-layer tellurenes, while the rest points are well aligned in line, independing on the parity of layers.
2) Lines 341-349, the Authors wrote: “As shown in Fig. 9, tuning of layers have strong effects on work functions of 2D-Te… …the work function of the 2D-Te decreases monotonically from 4.44 eV to 4.25 eV …Work functions shows noticeable change…” In my opinion, such variation is very small. The difference in these values (about 0.2 eV) calculated here doesn’t exceed the experimental error in a real experimental determination of a work function. Therefore, the fine tuning of the work function of tellur nanofilms using their thickness seems to be speculative.
3) What is the purpose for calculation of optical properties up to excitation energies of 20-35 eV? How correct is DFT in description of such highly excited states?
4) A slight polishing of English grammar is required (some plurals are used with “is”). Eq. (3) in line 254 has the not readable characters.
Reviewer 2 Report
This paper report a DFT investigation of the electronic and optical
properties of 1-layer and few-layers tellurene.
The paper is well written, computational methods appear to be adequate
and well justified, and results are scientifically sound.
I have only minor comments:
1) can the authors state which Te polymorphs is tellurene exfoliated from?
I think it's beta-Te, since alpha-Te is composed of single Te layers
with hexagonal symmetry.
2) are the optical properties calculated with Kohn-Sham orbitals, or
within the more sophisticated GW-BSE approach? then in the abstract
(line 19), "many body perturbation theory" should be removed.
3) line 118: "planner" => "planar"
Finally, I recommend publication of this paper after minor revisions.
Reviewer 3 Report
See the attached file
